# A Comprehensive Proteomic SWATH-MS Workflow for Profiling Blood Extracellular Vesicles: A New Avenue for Glioma Tumour Surveillance

**DOI:** 10.3390/ijms21134754

**Published:** 2020-07-03

**Authors:** Susannah Hallal, Ali Azimi, Heng Wei, Nicholas Ho, Maggie Yuk Ting Lee, Hao-Wen Sim, Joanne Sy, Brindha Shivalingam, Michael Edward Buckland, Kimberley Louise Alexander-Kaufman

**Affiliations:** 1Neurosurgery Department, Chris O’Brien Lifehouse, Camperdown 2050, Australia; susannah.hallal@lh.org.au (S.H.); brindha.shivalingam@lh.org.au (B.S.); 2Brainstorm Brain Cancer Research, Brain and Mind Centre, The University of Sydney, Camperdown 2050, Australia; heng.wei@sydney.edu.au (H.W.); nicholas.ho@sydney.edu.au (N.H.); maggie.lee@sydney.edu (M.Y.T.L.); michael.buckland@sydney.edu.au (M.E.B.); 3Discipline of Pathology, School of Medical Sciences, The University of Sydney, Sydney 2006, Australia; 4Neuropathology Department, Royal Prince Alfred Hospital, Camperdown 2050, Australia; joanne.sy@health.nsw.gov.au; 5Dermatology Department, School of Medical Sciences, The University of Sydney, Westmead 2145, Australia; ali.azimi@sydney.edu.au; 6Department of Medical Oncology, Chris O’Brien Lifehouse, Camperdown 2050, Australia; haowen.sim@lh.org.au; 7NHMRC Clinical Trials Centre, University of Sydney, Camperdown 2050, Australia; 8The Kinghorn Cancer Centre, St Vincent’s Hospital, Darlinghurst 2010, Australia

**Keywords:** glioblastoma, glioma, extracellular vesicle, liquid biospsy, SWATH, DIA, mass spectrometry

## Abstract

Improving outcomes for diffuse glioma patients requires methods that can accurately and sensitively monitor tumour activity and treatment response. Extracellular vesicles (EV) are membranous nanoparticles that can traverse the blood–brain-barrier, carrying oncogenic molecules into the circulation. Measuring clinically relevant glioma biomarkers cargoed in circulating EVs could revolutionise how glioma patients are managed. Despite their suitability for biomarker discovery, the co-isolation of highly abundant complex blood proteins has hindered comprehensive proteomic studies of circulating-EVs. Plasma-EVs isolated from pre-operative glioma grade II–IV patients (*n* = 41) and controls (*n* = 11) were sequenced by Sequential window acquisition of all theoretical fragment ion spectra mass spectrometry (SWATH-MS) and data extraction was performed by aligning against a custom 8662-protein library. Overall, 4054 proteins were measured in plasma-EVs. Differentially expressed proteins and putative circulating-EV markers were identified (adj. *p*-value < 0.05), including those reported in previous in-vitro and ex-vivo glioma-EV studies. Principal component analysis showed that plasma-EV protein profiles clustered according to glioma histological-subtype and grade, and plasma-EVs resampled from patients with recurrent tumour progression grouped with more aggressive glioma samples. The extensive plasma-EV proteome profiles achieved here highlight the potential for SWATH-MS to define circulating-EV biomarkers for objective blood-based measurements of glioma activity that could serve as ideal surrogate endpoints to assess tumour progression and allow more dynamic, patient-centred treatment protocols.

## 1. Introduction

Diffuse gliomas are the most common and devastating primary malignant brain tumours of adults, owing to their characteristic diffusely invasive growth patterns. The principal histologic subtypes, astrocytoma and oligodendrogliomas, are graded by severity according to a set of prescribed histological and molecular features, ranging from WHO grade II–IV (least to most severe), with both subtype and grade contributing to prognosis [1]. The most prominent distinguishing molecular feature of diffuse gliomas is the isocitrate dehydrogenase (*IDH*) mutational status which separates tumours into two broad categories. In general, *IDH*-wildtype (*IDH*-wt) astrocytic tumours arise as primary de novo glioblastomas (GBM; astrocytoma grade IV) and are the most frequent and aggressive manifestations of diffuse glioma. GBMs are almost universally fatal and carry a dismal median survival of 15 months [2]. The current standard-of-care for GBM patients involves debulking surgery to remove the main tumour mass, followed by concomitant radiotherapy and temozolomide (TMZ) chemotherapy (Stupp protocol) [3]. Unfortunately, treatments only provide temporary palliation to patients, and GBM recurrences are inevitable [4]. GBM tumours often re-emerge as evolved, treatment-resistant entities and death typically ensues within 6-months of recurrence [2]. *IDH* mutations confer a prognostic advantage and are diagnostic of grade II–III gliomas or secondary GBMs that have progressed from grade II–III astrocytomas [5,6]. Tumours with *IDH*-mutations (*IDH*-mut) are further subdivided according to 1p/19q codeletion status, where oligodendrogliomas harbour both *IDH*-mut and 1p/19q codeletion and carry the most favourable prognosis. Malignant progression of *IDH*-mut gliomas is also common, including the appearance of a ‘hypermutation phenotype’ following TMZ therapy [7].

Apart from a lack of targeted treatment strategies, a significant obstacle to effective clinical management of diffuse glioma is the dearth of sensitive approaches to monitor tumour progression and treatment-resistant recurrence. Histological evaluation of brain tissue is the only definitive method for diagnosing glioma progression and/or recurrence [8]. Yet, routine neurosurgical biopsies are impractical for tumour surveillance, contribute significantly to patient morbidity [9] and have inherent under-sampling issues due to the highly heterogenous nature of these tumours [10]. Neuroimaging and neurological assessments are used to monitor glioma tumours yet are insensitive to early signs of recurrence. Moreover, treatment-related effects such as pseudoprogression and radiation necrosis, share radiological features with tumour progression and can confound the detection of recurrent disease [11,12,13,14,15,16,17]. Generally, pseudoprogression is observed as a radiographic worsening of a tumour which is indistinguishable from true tumour progression [15] and occurs in around 36% of high-grade glioma patients [13]. Radionecrosis is a severe local inflammatory reaction in response to radiotherapy [16]. MRI features of radionecrosis and tumour recurrence overlap considerably [17], however, there is often a combination of both entities [18]. When radionecrosis is accompanied by neurological symptoms, managing the clinical care of glioma patients is particularly difficult. While revisions have been made to Response Assessment in Neuro-Oncology (RANO) criteria to standardise tumour monitoring [19], discerning treatment-related effects from true tumour growth remains a challenge.

Improving patient outcomes ultimately requires the development of sensitive methods that can accurately, efficiently, and sensitively monitor glioma activity and treatment response. With efforts to improve the clinical management of cancer comes a growing trend to design minimally-invasive liquid biopsies [20,21,22]. Liquid biopsies sample and measure tumour-derived factors or ‘biomarkers’ in body fluids, i.e., a blood test, to assess the presence or activity of a tumour in real-time [20]. Currently, no blood-based glioma biomarkers are in routine clinical use. Extracellular vesicles (EVs) are 30–1000 nm stable membranous particles that are released by all cell types and hold major promise for biomarker discovery research. EV secretion is increased in neoplasia [23,24] and glioma-derived EVs traverse the blood–brain-barrier and are detectable in the circulation [25,26], allowing glioma molecular information to be sampled from peripheral blood [27]. We previously showed that proteins cargoed in EVs surgically captured from tumour microenvironments can distinguish high-grade from low-grade glioma patients [28,29]. Sampling EVs from the circulation and screening their proteomic content may therefore serve as a complementary and minimally invasive approach to assess the glioma molecular landscape in real-time as tumours evolve [20].

Despite the suitability of circulating-EVs for biomarker discovery, comprehensive proteomic characterisation has been challenging. Glioma-EVs constitute only a minor subset of the total circulating-EV population [30]. While attempts have been made to capture and analyse circulating GBM-EVs by targeting common GBM cell-membrane proteins on the EV surface [31], no universal markers have been identified that can accurately and reproducibly target all populations of circulating GBM-EVs. Cell-membrane proteins are topologically reversed on the EV surface [32]; the unequivocal and reliable capture of circulating GBM-EVs is dependent on complete profiles of GBM-EV surface-accessible protein sequences which are not yet characterised. Analysing only a subset of circulating-EVs might also exclude any systemic effects of tumour pathology and/or radio-chemotherapy [33], that could otherwise indicate disease burden and treatment effectiveness.

Despite recent advances in EV isolation methods, recovering pure EV populations from blood is not yet possible. Blood is an exceptionally complex fluid, comprising extraordinarily diverse protein species with wide dynamic range in concentrations. The most abundant 22 proteins account for 99% of plasma proteins by weight [34]. High abundance blood proteins, including albumin, haemoglobin, serotransferrin, complements and immunoglobulins, commonly co-isolate with circulating EVs [35] and mask the detection of putative EV proteins using by traditional *shot-gun* tandem MS (MS/MS) analyses [36] using information dependent acquisition (IDA) methods [37]. The limitation of IDA in biomarker discovery of complex biological samples is due to its inherent dependence on inclusion lists which selects for highly abundant peptides to be assessed by MS, often leading to putative disease-related biomarkers going undetected, unidentified and neglected [36]. Strategies to selectively deplete high abundance proteins are fraught with limitations, mainly relating to reproducibility issues and the co-depletion of off-target proteins bound to large abundant blood transport proteins, like albumin (i.e., ‘the albuminome’).

More recent MS strategies cater to biomarker discovery of complex biological samples by using highly specific and targeted data independent acquisition (DIA) strategies. Sequential window acquisition of all theoretical mass spectra (SWATH) is a form of DIA on Sciex TripleTOF 5600+ instruments (Sciex, Framingham, MA, USA). SWATH is a label-free MS method that theoretically allows all peptides in a sample to be identified and quantified. The method was first described by Gillet et al. [38] and involves a targeted data extraction strategy that mines SWATH-MS data against an IDA spectral library [39]. The SWATH-MS method involves fragmenting and analysing all ionised peptides across SWATH windows of a specified mass-range in an unbiased fashion so that all ions undergo MS/MS, enabling sensitive and accurate quantitation, even for low abundant peptides [40,41]. High-resolution extracted ion chromatograms (XICs) are drawn for the fragment ions for every peptide in the sample [42]. The SWATH-MS data can then be aligned to a high-quality comprehensive spectral library containing MS coordinates of (i) the peptide precursor ion *m/z*, (ii) the *m/z* of the fragment ions and their intensities and (iii) the chromatographic retention time for each target peptide [42]. This information allows proteins that were detected by SWATH-MS to be identified and quantified if they are present in the library [42]. As such, SWATH-MS data can be archived and analysed retrospectively, allowing for maximal identifications as spectral libraries mature.

This study aimed to establish a method that would allow comprehensive proteomic profiling of circulating-EVs and to determine whether this approach could distinguish glioma subtypes and control cohorts. Here, a targeted SWATH-MS proteomic workflow was employed for the identification and quantification of plasma-EV proteins (Figure 1). A comprehensive glioma spectral library was developed and used to extract SWATH-MS data corresponding to proteins detected in circulating-EVs. As discoveries in circulating-EVs have immense potential for the development of new clinical tests, it was important to optimise our experimental workflow with an EV isolation method that isolates relatively pure EV populations in a rapid, efficient and scalable fashion so that future EV biomarker panel tests can be readily adopted by hospital pathology services. We show that SWATH-MS is a suitable and promising method for advancing biomarker discovery using EVs captured by size exclusion chromatography (SEC) from the plasma. Plasma-EV proteomes were able to stratify glioma patients and EVs re-sampled from patients with confirmed recurrent tumour progression showed protein changes consistent with more aggressive glioma-EV profiles. These exciting findings may be pivotal for the shift towards precision care models in the management of diffuse glioma.

## 2. Results

### 2.1. Characterisation of Plasma-EVs

EV elution and size distribution profiles for all 12 EV fractions were consistent across plasma sampled from three healthy individuals. A discernible EV population was first observed to elute in fraction 7, and the EV concentration and average particle to protein ratio increased for each subsequent fraction (Figure 2A). EV containing fractions 7–12 were predominantly comprised of small vesicle subtypes of smaller than 200 nm, with modal size distributions of 80–100 nm and a vesicular morphology (Figure 2B,C). Label-free LC-MS/MS sequencing of fractions 7–12 determined those most suitable for blood-based biomarker discovery, i.e., fractions with the highest coverage of EV-related proteins and lowest levels of soluble blood proteins. The highest number of unique protein IDs were detected in fractions 7–9 and corresponded to a higher number of confident EV-related protein identifications (Figure 2D and Appendix A). Functional enrichment analysis showed that plasma-EV fractions 7–12 had significant annotations to exosomes and other membranous compartments (*p* < 0.01), with higher enrichments for fractions 7–9 compared to 10–12 (Figure 2E). Overall, 82 of the top-100 canonical EV proteins, as reported by Vesiclepedia, were identified in the plasma-EV fractions; 93% (75/82) of these EV-related proteins were exclusively detected across fractions 7, 8, and 9 (Appendix A). The higher number of protein identifications in the earlier eluting EV fractions coincided with lower spectral counts for the common highly-abundant serum-related proteins, which increased with each sequential EV fraction (Figure 2F); there were increases in normalised total spectra for A2M (5.3-fold), C3 (3.4-fold), TF (3.3-fold), CFH (23.4-fold), HP (5.3-fold) and IGG1 (2.7-fold) in fraction 12, relative to fraction 7. As such, the earlier eluting plasma-EV fractions 7–9 were considered the most attractive candidates for large-scale proteomic analysis for EV biomarker discovery.

### 2.2. Assembling a Spectral Library Tailored to Glioma SWATH Analysis

IDA LC-MS/MS analysis of HILIC fractionated peptides isolated from glioma and other tumour specimens yielded a total of 10,528 identified protein species at an unused ProtScore detection threshold of 0.05. Of these proteins, 9414 had a maximum global FDR of 5% and were included in the spectral library. In the library, 8662 proteins were associated with at least two unique peptides and transitions, and were included in the Skyline target list for SWATH data extraction.

### 2.3. SWATH-MS Data Extraction, Quality and Reproducibility

Plasma-EV proteomes (total *n* = 52; 41 glioma II–IV, five non-glioma controls and six healthy controls; Table 1), were sequenced by SWATH-MS and the peaks were aligned to the spectral library. To ensure accurate assignment of the SWATH-MS peaks, the retention times of the SWATH-MS peaks were aligned to the IDA spectral library using an indexed retention time (iRT) calculator generated from 20 heavy-labelled *PepCalMix* reference peptides spiked and measured in both plasma-EV peptides and spectral library specimens. The iRT linear regression had a Pearson’s correlation coefficient (*r*^2^) of 0.9903, reflecting highly correlative measured retention times and *PepCalMix* iRT definition values (Appendix A). The retention times of the *PepCalMix* peptides were also highly reproducible across all plasma-EV samples (Appendix A), and normalised peak areas of the standard peptides were similar across the sample groups (Appendix A). This ensured that the plasma-EV peptides analysed by SWATH-MS could be reliably and reproducibly aligned to the spectral library for accurate quantitative comparisons. The SWATH-MS data quality was further assessed by manual inspection of randomly selected extracted ion chromatographs (XIC) for *PepCalMix* peptide, SPYVITGPGVVEYK, and Aurora Kinase A peptide, VLCPSNSSQR to assess peak shape, retention time stability, signal intensity and signal-to-noise ratio (Appendix A). All parameters were highly reproducible across samples, with identical peak shapes observed that had minimal noise and <1 min chromatographic retention-time shifts across samples.

### 2.4. SWATH-MS Based Characterisation of Glioma Plasma-EV Proteomes

Overall, 4909 EV-associated proteins were confidently aligned to the IDA spectral library for all 52 plasma samples (*q* ≤ 0.01). Grouped by genetic-histological subtype, the MSstats package confidently identified and measured 4834 proteins in *IDH*-wt GBM, 4814 in *IDH*-mut astrocytoma (AST), 4270 in *IDH*-mut oligodendroglioma (OLI), 4488 in grade I meningioma (MEN), and 4586 in healthy control (HC) cohorts (Appendix A. Similarly, when grouped by glioma grade II–IV (GII-IV), 4836 proteins were identified for GIV, 4750 in GIII and 4730 in GII (Appendix A). FunRich analysis of the plasma-EV proteins showed that all glioma subtypes and controls were similarly functionally annotated to significant biological processes (*p* < 0.01), such as cell growth and/or maintenance (7.9 ± 0.07%), transport (9.2 ± 0.08%) and metabolism (14.8 ± 0.15%). Interestingly, the plasma-EV proteins were also mapped to the brain (64.9 ± 0.35%) and malignant glioma (54.0 ± 0.33%), as significant ‘sites of expression’. A total of 4054 proteins were common to plasma-EVs from all sample cohorts, and 90% of these (3648 proteins) were annotated as ‘extracellular vesicle’, ‘exosome’, ‘microparticle’ and ‘microvesicle’ proteins in Vesiclepedia (Figure 3A). Unsupervised clustering by PCA of EV-associated protein levels showed distinct grouping of plasma samples from patients with highly malignant gliomas (GBM/GIV) and control cohorts (HC and MEN; Figure 3B-1,B-2), with less malignant glioma tumours intermediately dispersed (GII-III, Figure 3B-1; AST and OLI, Figure 3B-2).

### 2.5. Glioma Plasma-EV Protein Profiles

We identified 463 plasma-EV DE proteins between glioma genetic-histological subtypes (adj. *p*-*value* ≤ 0.05; Appendix A), and 318 DE proteins between glioma grades (adj. *p* ≤ 0.05; Appendix A). Gene ontology enrichment by PANTHER revealed that the significant DE plasma-EV proteins were predominantly associated with energy generation by mediating proton transport (6.06-fold enrichment, *p-value* = 0.0491), vesicle-mediated transport (2.12-fold, *p* = 0.022), carboxylic acid biosynthesis (4.99-fold, *p* = 0.0194) and protein folding (3.88-fold, *p* = 0.0352). Significant plasma-EV proteins were implicated to reactome pathways, including WNT5A-dependent internalisation of FZD4 (12.12-fold enrichment, *p-value* = 0.0435), EPH-Ephrin signalling (7.76-fold, *p* = 0.011), innate immunity (2.47-fold, *p* = 1.6 × 10^−7^) and vesicle-mediated transport (3.03-fold, *p* = 2.94 × 10^−8^; Appendix A). Ingenuity^®^ functional pathway analysis of DE proteins between GBM and HC plasma-EVs (463 proteins; FC ≥ 2, adj. *p-value* ≤ 0.05) revealed significant associations to cancer (*p-value* range = 2.28 × 10^−2^–8.10 × 10^−19^, 450 molecules) and clathrin-mediated endocytosis signalling (*p* = 1.37 × 10^−4^). The top scoring molecular and cellular functions included cellular function and maintenance (*p-value* range: 2.07 × 10^−2^–8.67 × 10^−9^, 104 molecules), protein synthesis (*p-value* range: 2.05 × 10^−2^–6.42 × 10^−5^, 80 molecules) and post-translational modifications (*p-value* range: 2.28 × 10^−2^–3.19 × 10^−4^, 24 molecules; Appendix A).

Distributions of the top-50 DE proteins across the genetic-histological subtypes and WHO grades II–IV are displayed as heat-maps (Figure 4A-1,A-2). PCA visualisation of sample clustering based on the expression of the top-50 proteins revealed a tendency for glioma samples to group away from controls and with their respective genetic-histological subtype (Figure 4B-1) and WHO grade (Figure 4B-2). Plasma EVs captured from glioma patients with the most aggressive glioma tumours (GBM/GIV) were highly distinguishable from controls, clustering to the far left of the PCA, while plasma-EVs derived from less aggressive tumours (AST and OLI, Figure 4B-1; GII-III, Figure 4B-2) were scattered between the GBM cohort and controls. Strikingly, the protein profiles of plasma-EV collected from three patients with recurrent tumour progression each show a shift towards a more aggressive glioma phenotype on the PCA plots. The EV protein profile of a grade II *IDH*-mut astrocytoma patient that progressed to grade III shifted to the left of the PCA, closer to the grade IV GBM samples (circle: AST, Figure 4B-1; GII to GIII, Figure 4B-2). A grade III *IDH*-mut astrocytoma plasma sample that progressed to grade IV (star: AST, Figure 4B-1; GIII to GIV, Figure 4B-2) also shifted to the left and grouped with the bulk of grade IV GBM tumours. Lastly, plasma sampled from a patient with a *IDH*-wt GBM recurrence (rectangle: GBM, Figure 4B-1; GIV, Figure 4B-2) also had an observable shift to the left of the PCA, conforming to this proposed trend of increased glioma aggression.

### 2.6. Putative Plasma-EV Biomarkers for Distinguishing Glioma Subtypes

The SWATH-MS data was further interrogated to identify plasma EV-associated proteins with expression restricted to the different glioma subtypes. Eleven proteins were exclusively found in plasma-EVs from GBM patients, i.e., AIDA, ARHGEF10, BNIP3L, FYB1, KMT2D, MAP7, MAST4, PDE8A, POLR2D, RENBP and SLC25A17 (Figure 5A). CDC40 was restricted to AST plasma-EVs and TPST2 in OLI plasma-EVs alone. Correspondingly, the 11 GBM plasma-EV proteins were also restricted to GIV samples, in addition to CETN3, PPP1R11 and SYT7. CDC40 and TPST2, detected in AST and OLI samples, respectively, were exclusively expressed in plasma from GIII patients (Figure 5A). No proteins were specific to GII patients or to the control cohorts. Proteins restricted to EVs from GBM/GIV patient plasma were further investigated in silico to establish whether their presence in circulating EVs reflects GBM tumour molecular pathophysiology and/or is consequent to EV selective packaging. Transcript levels corresponding to the GBM/GIV restricted proteins were generated by Oncomine (Compendia Biosciences, MI, USA) using TCGA data (GBM, *n* = 542; normal brain, *n* = 10; Human Genome U133A Array, TCGA 2013 [43]). Significantly higher transcript levels were observed in GBM tumours for AIDA (2.5-fold, *p* = 3.4 × 10^−8^), BNIP3L (1.2-fold, *p* = 5.5 × 10^−6^), CETN3 (1.8-fold, *p* = 1.7 × 10^−7^), FYB1 (1.8-fold, *p* = 7.25 × 10^−9^) and POLR2D (1.7-fold, *p* = 8.14 × 10^−5^; Figure 5B).

DE analysis of glioma plasma EV proteins (categorised by genetic-histologic subtype and WHO grade) relative to HC and MEN (non-glioma control) were identified (adj. *p* ≤ 0.05, FC ≥ 2) and are tabulated in Appendix A. To assess whether the plasma EV proteomes can distinguish glioma from non-glioma patients, proteins with significant expression relative to both control groups (HC and MEN) were collated. Significant, DE ‘glioma-associated proteins’, including 80 GBM, 12 AST and 14 OLI proteins, are displayed in a Venn diagram along with their direction of change relative to both control groups (Figure 5C-1) and are listed in Appendix A. Of these proteins, EBNA1BP2 and FAM129A were significantly expressed in the GBM, AST and OLI samples relative to controls. Similarly, when grouped by WHO grade, 78 GIV, three GIII, and eight GII proteins were significant relative to both controls, (Figure 5C-2; Appendix A). Here, AK2, CYB5A and GOLT1B were significantly higher in all glioma sample groups, relative to controls.

### 2.7. Comparisons to Previous GBM-EV Proteomics Studies

The plasma-EV SWATH-MS data was interrogated for previously reported in vitro ‘GBM-EV signature proteins’ common to EVs released by six established GBM cell-lines [29]. We measured 130 of the 145 ‘signature’ proteins (90% overlap) in circulating plasma-EVs from glioma patients. Of these, 10 EV-associated proteins were significantly DE in GBM patient plasma, including PSAP, CALR, PLOD3, HSPA4, GANAB, LGALS3BP, CCT2, PPIA, C3 and KRT10 (Figure 6A). We then compared the plasma EV-associated proteins resolved here to the 298 proteins identified in EVs captured from neurosurgical aspirates from glioma grade II–IV tumour resection surgeries [28], and observed an 87% overlap (260/298 proteins) between the two studies. Twelve of these neurosurgical aspirate EV proteins also showed significance in GBM plasma-EVs, including ANXA2, UQCRC2, COX5A, NDUFS4, IARS, TMEM65, CCT7, PSMC3, CCT2, GABBR1, SCD5 and TOMM40 (Figure 6A). From this list of previously reported GBM-EV proteins, PSAP, PPIA, CCT7 and C3 were consistently significantly DE in GBM/GIV plasma-EVs relative to both control groups (Figure 5A). PSAP (Figure 6B-1), PPIA (Figure 6B-2), CCT7 (Figure 6B-3) and C3 (Figure 6B-4), are displayed as boxplots to show the distribution of their expression across the plasma-EVs of patients.

## 3. Discussion

Accurate and sensitive glioma blood-based biomarkers will arm clinicians with much-needed information and offer patients more definitive and timely answers about the active state of their cancer and the effectiveness of their treatment. Circulating-EVs hold real promise as robust and readily accessible pools of glioma biomarkers. While high-throughput next-generation-sequencing technologies have allowed thorough characterisation of nucleic acids in blood-derived EVs [27,44], the proteome of circulating-EVs remains largely unexplored. Shot-gun proteomic methods have been primarily used to investigate glioma-EV proteomes [45,46]. However, such approaches achieve limited proteomic coverage, especially for complex blood-derived specimens [47]. We have employed SWATH-MS in conjunction with a data extraction strategy against a comprehensive spectral library for in-depth assessment of the proteomic content of circulating-EVs from glioma patients. The expansive coverage of targeted SWATH-MS allowed the identification of potential peripheral EV-associated glioma biomarkers, as well as the translation of previous in vitro [29] and ex vivo findings [28]. Multiple characteristic EV proteins (68 proteins, including ACTB, ACTN1/4, ANXA1, ANXA2, ANXA6, CD9, FLOT1, GAPDH, HIST1H4A, HSP90AB1, ITGA6, ITGA2B, PDCD6IP, SLC3A2; see Appendix A) and glioma-associated proteins (including CCT7, PPIA and C3; Appendix A) were confidently identified in plasma-EVs by both IDA and SWATH (DIA) MS methods. The development of assays using targeted DIA strategies (i.e., selective/parallel reaction monitoring) or immunoassays will be necessary to validate the putative glioma EV markers resolved here.

In the literature, there are no definitive propositions for the use of plasma over sera for EV biomarker discovery. The process to separate serum from blood first requires blood coagulation, which causes platelet activation and leads to increased platelet-EV secretion [48]. Sera also comprises a higher proportion of particles larger than 200 nm, compared to plasma [49]. Thus, plasma was selected as the starting material to optimise our biomarker workflow here. SEC was chosen to isolate plasma-EVs for discovery proteomic analysis as it is rapid and reliable for isolation of small-EV subtypes from complex biological fluids. While SEC does retain soluble protein and lipoprotein contaminants in EV preparations [50,51], it is readily adaptable, scalable and automatable, making it ideal for routine EV isolation in a diagnostic pathology service environment. The suitability of our discovery plasma-EV proteomics approach was evaluated using SEC fractions from healthy plasma (*n* = 3) and the isolated EV populations were characterised to meet the minimal information for studies of extracellular vesicles 2018 (MISEV 2018) criteria [52]. The presence of canonical EV proteins and limited serum protein contaminants (Figure 2) determined that plasma-EV fractions 7–9 were the most suitable for large-scale proteomic analysis.

### 3.1. SWATH Produces High-Quality Proteomic Data for Plasma-EV Biomarker Discovery

A comprehensive custom 8662-protein spectral library was constructed using fractionated peptides derived from multiple glioma specimens and other cancer-related material to allow for maximal targeted identification and relative quantitation of plasma-EV proteins. The spectral library contained peptides from GBM cells and tissues, as well as, EV peptides from glioma neurosurgical aspirates, for which previous proteomic analyses have confidently identified up to 3000 proteins [28]. Here, we exploited HILIC fractionation of the neurosurgical aspirate EV peptides to improve their proteomic coverage, and maximise the detection of glioma EV-related proteins sequenced by SWATH-MS. Circulating-EV peptides from diffuse glioma patients and controls were prepared and analysed by SWATH-MS. A targeted extraction method was used to align SWATH-MS retention times to the custom spectral library. The retention-time alignment was facilitated by *PepCalMix* standard peptides, spiked into both the library and plasma-EV specimens. Using an mProphet scoring algorithm and FDR estimation, this proteomic workflow facilitates confident and reliable peptide quantification across the plasma-EV biological replicates [53], with highly reproducible retention-times and XIC peptide peaks across all plasma-EV samples (Appendix A and Figure 5). Overall, we achieved a total of 4054 high-confidence protein identifications across the plasma-EV sample groups; the identified proteins had significant functional associations to exosomes and a high degree of overlap with Vesiclepedia-listed EV proteins (Figure 3A). These results demonstrate the capability of state-of-the-art SWATH-MS technology for circulating-EV proteome characterisation, allowing in-depth proteomic assessments of complex specimens while maintaining a high level of precision, reproducibility and accuracy across samples. While SWATH-MS sequences all peptides in a sample, in-depth protein identification is constrained by the extensiveness of the spectral library used. As spectral libraries mature, plasma-EV samples can be re-analysed to achieve more complete protein profiles, however, analyses of large SWATH datasets are both time and computationally intensive.

### 3.2. Plasma-EV Proteomes Reflect Glioma Subtype and Aggression

The value of SWATH-MS as a biomarker discovery platform was further demonstrated by the observed clustering of glioma subtypes based on plasma-EV protein expression levels, with plasma-EV profiles from diffuse glioma patients readily discernible from healthy and non-glioma controls (Figure 3B and Figure 4B). Moreover, EV profiles appeared to align along a trend of increasing glioma aggression, with GBM/GIV specimens clustering the furthest away from control samples and the less malignant subtypes (AST, OLI or GII-III) intermedially dispersed. Generally, OLI carries a more favourable prognosis than AST [54], and here OLI plasma-EV profiles were not only distinguishable from AST but were grouped in closer proximity to the control samples, while AST profiles were more similar to those of the GBM plasma-EVs (Figure 3B-1 and Figure 4B-1). Interestingly, resampled pre-operative plasma specimens from three patients with confirmed tumour progression exhibited EV profiles that shifted towards those with more ‘aggressive’ disease (Figure 4B-1,B-2). This observation highlights the potential of sampling blood-EVs as a bona fide approach to monitor glioma patients for recurrent progression.

While *IDH*-mut status can indicate glioma severity, GBMs were not readily distinguishable on the PCA plots by *IDH*-mut status (i.e., *IDH*-wt GBM vs. *IDH*-mut GBM). There were limited specimens from patients with secondary GBM studied; thus, significant plasma-EV proteins between the *IDH*-wt GBM and *IDH*-mut AST categories most likely reflect differences in WHO grade (i.e., GIV vs. GII-III). Four proteins were identified with significantly lower levels in *IDH*-wt GBM relative to *IDH*-mut AST, i.e., TIPIN (−50-fold), NUF2 (−16.7-fold), SMARCE1 (−10-fold) and PIEZO1 (−2.4-fold; Appendix A), and the utility of these discriminating protein markers should be further investigated. Interestingly, AIDA, BNIP3L, CETN3, FYB1 and POLR2D were specific to GBM plasma-EV, and correspondingly, significantly higher gene expression levels were observed in GBM tumour tissue relative to healthy brain (Figure 5B). AIDA, CETN3, FYB1 and POLR2D are also unfavourable prognostic markers across multiple cancer types [55]. Overall, these results indicate that plasma-EV protein profiling is capable of not only distinguishing between glioma patients, but identifying those with more aggressive disease. This might be rationalised by the larger quantities of tumour-derived EVs containing more oncogenic protein species that are secreted into the circulation [29,56,57] by more infiltrative, proliferative, higher-grade tumours [58].

### 3.3. Links to Previous GBM-EV Studies

Multiple protein species resolved in GBM/GIV plasma-EVs here were previously identified in in vitro [29] and ex vivo [28] GBM-EV studies (Figure 6A). Notably, PPIA, PSAP, CCT7 and C3 were significantly DE in GBM plasma-EVs, relative to controls (Figure 6B-1–B-4 and Appendix A). PPIA, also known as cyclophilin A, has roles across a range of biological processes, with essential functions in protein folding and chaperone activity [59]. PPIA overexpression is reported in various cancer types, its expression influenced by chemotherapy [60,61] and PPIA is a reported target gene of dysregulated miRNA species present in GBM patient plasma [62]. C3 is a member of the human complement system with key functions in innate immunity. C3 deposits in GBM tissue, suggesting a role of the complement cascade in GBM pathogenesis [63]. PSAP is a highly conserved glycoprotein and precursor of lysosomal proteins (saposins A–D) that is integral to the intracellular degradation of sphingolipids [64]. Previously, we showed that PSAP levels in GBM-EVs have a significant, positive correlation to in vitro GBM cell invasion [29] and multiple studies have reported high PSAP expression in clinical glioma specimens, glioma-stem cells and cell-lines [65,66]. PSAP was not confidently identified in EVs captured directly from diffuse glioma (II–IV) microenvironments [24,25], and here we report significantly lower PSAP levels in GBM plasma-EVs relative to controls. Paradoxically, high PSAP levels are associated with weakly- or non-metastatic prostate cancers, a reduction in distant metastasis [67] and PSAP serum levels are elevated in advanced-stage prostate cancer patients [68]. Although gliomas rarely metastasise beyond the CNS [69], reduced PSAP levels in plasma-EVs might reflect a mechanism whereby PSAP is selectively retained by GBM tumours to promote growth and invasion in the brain.

Interestingly, TOMM40 was measured in significantly high levels in plasma-EVs from GBM/GIV patient groups relative to HC (adj. *p* ≤ 0.05; Figure 5A, Appendix A). TOMM40 was also measured in higher levels in GIV, relative to GIII (adj. *p* ≤ 0.05) and GII (borderline significance of adj. *p* = 0.06; Appendix A). We previously reported significantly higher TOMM40 expression in EVs from GBM neurosurgical aspirates compared to GII-III glioma [28]. TOMM40 plays a central role in cell metabolism, shuttling proteins and acting as the gateway for protein entrance into the mitochondria [70]. High TOMM40 levels are associated with a super-invasive subtype of melanoma [70], with BRCA1/2 expression and mutations [71] and is a histological feature for the squamous subtype of non-small cell lung cancer [72]. Increased levels of TOMM40 in GBM/GIV plasma-EVs relative to GII-GIII gliomas, as well as non-glioma and healthy controls, may be a useful distinguishing circulating biomarker for GBM and should be further explored.

### 3.4. T-Complex Protein 1 Ring Complex (TRiC) Interactome May Indicate GBM Presence and/or Progression

T-complex protein 1 ring complex (TRiC; also known as chaperonin containing TCP1 complex, CCT) protein subunits, CCT2, CCT3, CCT4, CCT5, CCT7 and TCP1 were measured in high levels (FC ≥ 2) in plasma-EVs from GBM patients relative to the controls, with significant increases detected for CCT2 and CCT7 (adj. *p* ≤ 0.05, Appendix A). We previously reported significantly higher CCT2 and CCT7 levels in GBM neurosurgical aspirate-EVs compared to GII-III glioma, with analogous transcript expression levels and DNA copy numbers reported for CCT2 and CCT7 in GBM tissue compared to normal brain in TCGA datasets [28]. While the changes observed here are relative to healthy and non-glioma controls, the significantly high expression of TRiC components, CCT2 and CCT7, within plasma-EVs of GBM/GIV patients, not observed for less aggressive glioma subtypes, might therefore relate to primary GBM biology (*IDH*-wt) and disease severity.

TRiC is a torus-shaped complex composed of eight distinct non-identical protein subunits (TCP1, CCT2, CCT3, CCT4, CCT5, CCT6A, CCT7, CCT8) [73,74]. It is a cytoplasmic molecular chaperone that assists newly formed proteins to fold correctly [73,75]. Increases in TRiC expression have been associated with tumorigenesis and influencing molecular pathways that contribute to tumour progression, such as p53 and STAT3 [73,76,77,78]. Defects in TRiC activity are associated with protein misfolding and cytotoxicity [79] and have been observed in multiple neuropathologies [80,81,82]. The role of TRiC in tumour aggression has been further corroborated by the successful implementation of cytotoxic peptides such as CT20p that reduce tumour burden by targeting CCT2 [83]. CT20p inhibition of CCT2 reduces TRiC activity and alters the level of other TRiC components, interfering with the proper folding of oncogenic molecules such as STAT3 [83]. While TRiC is a highly functional chaperone as a complex, the subunits also have individual functions [84]. CCT7 expression is associated with cancer cell growth and maintenance [85], is downregulated during the DNA damage response in GBM [85] and is associated with poor clinical outcome in GBM patients [86]. CCT2 also has important functional links to cancer, including survival of breast cancer cells and negative survival associations in breast cancer [87], non-small cell lung cancer [88] and diffuse intrinsic pontine glioma [89].

TRiC’s folding activity requires assistance from other protein molecules that together comprise the TRiC interactome [75]. TRiC interactome proteins, PARK7 and RECQL, were also DE in plasma-EVs in this study. PARK7 was significantly increased in both GBM and GIV sample groups, relative to controls (Appendix A) and RECQL was a top-50 DE plasma-EV protein, with high levels distinguishing GIV from GIII gliomas (10.7-fold, adj. *p-val* = 0.0157, Appendix A) as well as GBM/GIV from controls. PARK7 has important roles in protecting cells from stress and apoptosis by acting as an oxidative stress sensor and redox-sensitive chaperone [90] and is increasingly implicated in neurodegenerative disease [91] and across multiple cancers, promoting tumorigenesis, stimulating cell proliferation, cell invasion and cancer metastasis [92,93,94]. PARK7 has been found in high concentrations in body fluids of cancer patients suggesting its potential as a non-invasive cancer biomarker [95]. In GBM, high cytoplasmic PARK7 levels are associated with strong nuclear p53 expression and inversely correlated with EGFR expression [92]. The RecQ helicase family is a group of highly conserved DNA unwinding enzymes involved in maintaining chromosomal instability [96]. RECQL-deficient cells have been shown to accumulate DNA damage, display sensitivity to DNA damaging agents and exhibit chromosomal instability [96,97]. RECQL is amplified and overexpressed in many clinical cancer specimens, and is associated with cell migration, invasion and metastasis, as well as exhibiting predictive and prognostic biomarker potential [96]. RECQL overexpression is reported in GBM [98] and RECQL has also been identified in cell-derived memetic nanovesicles and exosomes derived from neuroblastoma [99]. Taken together, results here and in previous GBM-EV studies [28,29], implicate a role for TRiC and its interactome in GBM pathology, which are reflected in EVs released into the circulation. It is possible that EV-associated TRiC and its interacting proteins could serve as robust circulating biomarkers for glioma and its progression. Further investigation of TRiC components and its interactome in plasma-EVs is warranted.

## 4. Materials and Methods

### 4.1. Pre-Operative Plasma Specimens and Patient Cohorts

Pre-operative bloods were collected from patients prospectively over a 36-month period. All patients provided research consent (Royal Prince Alfred Hospital Neuropathology Tumour and Tissue Bank, SLHD HREC protocol X19-0010) and specimens were processed and analysed under approved University of Sydney HREC project 2012/1684. Diagnoses were made by neuropathological assessment of surgically resected tumour material. Bloods were sampled from patients with primary GBM (grade IV *IDH*-wt astrocytoma, *n* = 24), secondary GBM (grade IV *IDH*-mut astrocytoma, *n* = 2), glioma grade II–III (*IDH*-mut astrocytoma (AST), *n* = 12; *IDH*-mut, 1p19q codeleted oligodendroglioma (OLI; *n* = 4)) and meningioma grade I (MEN, *n* = 5; non-glioma control). Bloods were re-sampled (prior to repeat surgeries) from three individual glioma patients with histologically confirmed tumour recurrence. Matching, re-sampled blood was captured from a patient with (1) progression from grade II to grade III *IDH*-mut astrocytoma; (2) progression from grade III to grade IV *IDH*-mut astrocytoma; and (3) recurrence of primary *IDH*-wt GBM. Plasma from healthy age-gender matched controls (HC, *n* = 6; three males, three females, age range 38–55 years) were also included. A summary of sample cohorts used for comparative analyses is provided in Table 1. More detailed patient demographics and diagnoses are provided in Appendix A.

### 4.2. Plasma Processing and EV Isolation

An overview of the experimental workflow is presented in Figure 1. Peripheral blood (15 mL) was collected into BD Vacutainer K2 EDTA tubes containing spray-dried K2 EDTA and processed within 4 h. The plasma was separated by a Ficoll-histopaque gradient (400× *g*, 30 min, RT; no brake). Platelets were depleted from the plasma by centrifugation (3000× *g*, 20 min, 4 °C) and 1.5 mL plasma aliquots were snap frozen in liquid N_2_ and stored at −80 °C.

Platelet-free plasma was thawed slowly on ice and EVs were purified using iZON qEV_original_ columns as per manufacturer’s instructions. Briefly, qEV columns were equilibrated with 5 mL PBS before loading 500 µL of platelet-free plasma onto the column. When the plasma passed through the reservoir, twelve 500-µL fractions were eluted in PBS. While each column can perform up to five EV isolations with PBS washes in between samples, only one column was used per specimen to circumvent signal carryover or cross-contamination. The plasma-EV fractions 1–12 were stored at −80 °C.

### 4.3. EV Characterisation by Nanoparticle Tracking Analysis, Transmission Electron Microscopy and LC-MS/MS

The size distributions and concentrations of the plasma-EV fractions were measured by nanoparticle tracking analysis (NTA) software (version 3.0) in triplicate using the NanoSight LM10-HS (NanoSight Ltd., Amesbury, UK), configured with a tuned 532 nm laser and a digital camera system (CMOS trigger camera). EVs were diluted with sterile-filtered PBS (viscosity 1.09 cP) to ensure 20–100 particles were detectable within the field of view of the standard CCD camera of the microscope. The NTA software captured triplicate 60 s video recordings of the EVs, at 25 frames per second with default minimal expected particle size, minimum track length, blur setting, with the temperature of the laser unit controlled to 25 °C. The videos were analysed by NTA3.0, which translates the Brownian motion and light scatter properties of each individual laser-illuminated particle into a size distribution (ranging from 10 to 1000 nm) and concentration (particles per mm) while simultaneously calculating their diameter using statistical methods [100]. The EV size distributions and concentrations were analysed in Microsoft Excel^®^. Plasma-EVs were also analysed by transmission electron microscopy (TEM). EVs were re-suspended in dH_2_O, loaded onto carbon-coated, 200 mesh Cu formvar grids (ProSciTech Pty Ltd.; Cat No. GSCU200C) and fixed with 2.5% glutaraldehyde in 0.1 M phosphate buffer (pH 7.4). Samples were negatively stained with 2% uranyl acetate for 2 min, dried at RT for 3 h and then visualised at 40,000×, 80,000× and 100,000× magnification on a Philips CM10 Biofilter TEM (FEI Company, OR, USA) equipped with an AMT camera system (Advanced Microscopy Techniques, Corp., MA, USA) at an acceleration voltage of 80–120 kV. Total protein content of the EV fractions were determined by a Qubit^®^ Protein Assay (Invitrogen, Carlsbad, CA, USA). The presence of canonical EV-marker and serum proteins in the plasma-EV fractions was confirmed by liquid chromatography tandem mass spectrometry (LC-MS/MS) and data was analysed in Scaffold Proteome Software, as described before [28].

### 4.4. Preparation of EV Proteomes for LC-MS/MS

The volume of plasma-EV fractions was reduced by vacuum centrifugation to 100 µL and EV proteins precipitated using chloroform-methanol. The precipitated EV proteins were dissolved in 90% (*v/v*) formic acid (FA) and dried by vacuum centrifugation. The dried EV proteins were resuspended in 0.2% (*w/v*) Rapigest SF^TM^ (Waters, Milford, MA, USA) in 0.05 mol/L triethylammonium bicarbonate (TEAB) and incubated at 95 °C for 5 min. Samples were sonicated twice with a step-tip probe at 30% intensity for 30 s to aid protein resuspension. Debris was removed by centrifugation at 13,000 rpm for 20 min at 4 °C. Proteins were then reduced in 12 mM Tris (2-carboxyethyl) phosphine (TCEP) (60 °C, 30 min) and alkylated in 50 mM iodoacetamide (RT, 30 min, in the dark). Sample pH was adjusted to 7.5 with 0.05 M TEAB and proteins were digested overnight at 37 °C with sequencing-grade trypsin (Promega, Madison, WI) in 1:30 (*w/w*) trypsin:protein ratio. The peptide solution was adjusted to a pH < 2 using 50% (*v/v*) FA and incubated for 30 min at 37 °C to cleave the Rapigest SF^TM^ detergent, which was then removed by centrifugation (16,000× *g*, 10 min, 4 °C). Peptides were then desalted by solid-phase extraction using 1cc HLB cartridges (Waters, MA, USA) and eluted in 70% acetonitrile (ACN)/0.1% FA (*v/v*). Protein and peptide concentrations were measured in duplicate using the Qubit^®^ Protein Assay Kit (Invitrogen, Carlsbad, CA, USA). Peptides were dried by vacuum centrifugation in 25 μg aliquots. The dried, ‘clean’ peptide mixtures were stored at −80 °C.

### 4.5. Generating a Custom, Comprehensive Glioma Spectral Library

LC-MS/MS analysis of fractionated GBM samples: a range of GBM samples (EVs, cells and tumour tissue fractions; see Appendix A for details) were used to create a comprehensive, custom glioma IDA spectral library. Specimens were prepared as described before from neurosurgical aspirate EVs [28], patient-derived GBM-stem-like cells [101], and soluble and microsomal proteomes [102] from GBM tumour tissue. Peptide samples were fractionated by reverse-phase hydrophilic interaction liquid chromatography (HILIC) to reduce sample complexity to increase LC-MS/MS protein ID coverage [103]. Briefly, desalted pooled peptides (20 µg) were dried by vacuum centrifugation, resuspended to 0.5 µg/µL in 90% ACN/0.1% trifluoroacetic acid (TFA) and fractionated with an Agilent 1200 high performance liquid chromatography (HPLC) system (Agilent Technologies, Santa Clara, CA, USA) on an in-house 17 cm TSKgel Amide-80 HILIC column (3 µm particle size) with a post-column PEEK^TM^ filter (Upchurch Scientific, Rohnert Park, CA, USA). Mobile phase buffers were comprised of A: 0.1% (*v/v*) TFA (Sigma, Chromasolv HPLC grade, Cat.No. 34851-4) and B: 90% (*v/v*) ACN/0.1% (*v/v*) TFA. The samples were loaded in buffer B and eluted over a 50 min gradient comprised of 100% B for 10.5 min (flow rate of 10 µL/min), followed by 100–60% B in 26.5 min, 60–30% B in 4 min, 30% B for 1 min and 100% A for the rest of the gradient at a flow rate of 6 µL/min. Chromatographic performance was monitored at 210 nm. Uniform peptide fractions were eluted in buffer B between 13 and 35 min. The number of HILIC fractions collected is specified in Appendix A. The resultant peptide fractions were dried by vacuum centrifugation. LC-MS/MS data acquisition was performed on an AB Sciex TripleTOF^®^ 6600 Quadrupole Time-Of-Flight (QTOF) mass analyser. The TripleTOF^®^ system was operated in IDA mode to acquire the spectral library data for HILIC fractionated peptides (Appendix A). The IDA method involved a MS survey scan range of 350–1700 m/z (0.25 s accumulation time) in positive ionisation mode, with rolling collision energy and dynamic accumulation selected. MS/MS scans were acquired in high sensitivity mode covering a mass range of 100–2000 m/z (25 ms accumulation time) of the 20 most intense ions with charge states of 2+ to 5+ and a mass tolerance of 20 ppm. Mass-charge ratios selected for MS/MS were dynamically excluded for 10 s. Prior to loading the samples, a LC-MS/MS standard consisting of 30 fmol pre-digested BSA was injected to test the performance and dynamic range of the instrument.

To generate our comprehensive glioma spectral library, LC-MS/MS data from HILIC fractionated GBM specimens were added to IDA spectral data captured from a range of other cancer specimens, acquired under the same conditions [104]. Protein database searching was performed for all fractions against a reference Human Uniprot FASTA database (3 October 2018) by ProteinPilot^TM^ Software 5.0 (SCIEX; Framingham, MA, USA) using a Paragon^TM^ algorithm. ProteinPilot search parameters included Cys alkylation-iodoacetamide; digestion-trypsin; instrument-TripleTOF 6600; species-homo sapiens; detection protein threshold of 10% and false discovery analysis. Identification focus was set to biological modifications and the search effort was thorough. The resulting spectral library file (group; deposited to PeptideAtlas, ID: PASS01487) was imported to Skyline 4.2 (Maccoss Lab, University of Washington, Seattle, WA, USA) and only peptides with a confidence score >0.05 were included in the spectral library.

### 4.6. SWATH-MS Data Capture and Peak Extraction

Peptide mixtures prepared from patient plasma-EVs (Table 1) were analysed by TripleTOF^®^ system operated in SWATH mode and covered a total of 159 custom, variably-sized windows (with a 1.0 Da window overlap) over a precursor mass range of 350–1750 *m/z* (Appendix A). Dried peptides were resuspended in 3% ACN (*v/v*)/0.1% (*v/v*) FA to a concentration of 0.4 µg/µL and *PepCalMix* heavy-labelled peptides (SCIEX) were added to each fraction to a final concentration of 6.67 fmol/µL. The resuspended peptides (2 µg) were injected onto an in-house 15 cm C_18_ reversed-phase column (75 µm diameter and 5 µm particle size) and analysed by the TripleTOF^®^6600 coupled to an Eskpert^TM^ NanoLC 425. The HPLC solvent system was comprised of buffer A: 0.1% (*v/v*) FA (Thermo Scientific, Cat.No. 85178) and buffer B: 80% (*v/v*) ACN (Thermo OPTIMA LC/MS grade, Cat.No. 34851-4), 0.1% (*v/v*) FA. Peptides were eluted over a 120 min gradient (2–35% B for 80 min, 35–95% B for 19 min, 95% B for 5 min and 2% B for 16 min).

Chromatographic peaks for the plasma-EV samples were extracted by aligning the SWATH-MS acquisitions to the comprehensive spectral library in Skyline 4.2 (Maccoss Labs). Before importing the SWATH-MS data to Skyline, an isolation scheme (Appendix A) was created according to the 159 variable isolation windows (Appendix A) used to acquire SWATH-MS data on the TripleTOF^®^ 6600. Skyline peptide and transition settings were prepared for SWATH-MS analysis as illustrated (Appendix A) and the background proteome was created by the human uniprot protein database (3-10-2018). Briefly, Skyline was configured to filter full-scan precursor ions (MS1) by count (three isotope peaks at a resolving power of 30,000 with a Time of Flight (TOF) mass analyser). Tandem MS (MS2) was filtered for DIA, TOF and a resolving power of 30,000. Transitions were filtered for precursor charges of 1+, 2+, 3+ or 4+, ion charges of 1+ or 2+, and ion types *y*, *b* or *p*. The MS/MS spectral library was added to Skyline, filtered for spectra with *q* ≤ 0.05, and peptides that matched multiple proteins, did not match any proteins, or did not meet the filter settings were removed. The raw SWATH-MS (.wiff) acquisitions of plasma-EV samples were imported to Skyline and aligned to the spectral library using an indexed retention time (iRT) calculator. The iRT calculator was created and calibrated to the retention times for the heavy-labelled *PepCalMix* peptides that were spiked-in to the spectral library (Appendix A). The SWATH-MS peaks were used if their retention time was within a 5 min window of their predicted retention time. The mProhpet algorithm was applied to the SWATH data [105] to quantify the confidence that a peak corresponds to its targeted peptide. Peaks with *q* < 0.05 were extracted and integrated for analysis. The peaks were also assessed for quality and analysed if at least half of the transitions contributed a co-eluting peak. FDR control was estimated by *q*-values, where peptides with *q* ≤ 0.05 (FDR 5%) were selected for label-free quantitation. The reproducibility and reliability of the SWATH-MS data and the performance of the mass spectrometer were assessed by manual inspection of randomly selected peptides for peak shape, retention time stability and signal intensity (Appendix A). MSstats input files were generated for differential expression (DE) analysis of the proteins in R.

### 4.7. Differential Expression (DE) Analysis and Statistics

Group comparisons were performed for identified proteins with MSstats3.7.3, an open source R package in Bioconductor [106]. Prior to DE analyses, biological replicates were annotated into their respective cohorts in Skyline 4.2. MSstats Input reports were generated and analysed in RStudio 1.0.153 and R3.5.0 using MSstats3.7.3. Briefly, the input reports were pre-processed by removing iRT proteins, allocating NAs for truncated peaks and assigning 0 for intensities with a detection *q*-value < 0.01. The data was then processed using the default MSstats settings which involved log_2_ transformation, normalisation by equalising the median intensities across runs, model-based quantification by Tukey’s median polish (TMP) and imputation of censored missing values using the accelerated failure time (AFT) model. Group comparisons were performed to yield fold change (FC), log_2_ fold change (log_2_FC), standard error of the log_2_ fold change (SElog_2_FC), Student’s *t*-test *p*-value, and Benjamini–Hochberg corrected *p*-values with a 5% FDR threshold.

Principal component analysis (PCA) was performed in R to visualise sample clustering. The top-50 DE significant proteins across all comparisons were identified by filtering and ranking proteins by their adjusted *p*-value and FC. Significant proteins were visualised with heat-maps plotted in R. Heat-maps of the top-50 significant proteins were plotted as log_2_-transformed FC, calculated by subtracting the protein abundance from its mean value for that protein across all samples. Pathway analysis was performed using Ingenuity^®^ software (Ingenuity Systems, USA; default core expression analysis) to assess functional associations (biological and canonical pathways) of DE plasma-EV proteins between GBM and healthy controls. The PANTHER over-representation test was used to perform gene ontology enrichment analysis of biological functions and reactome pathways for the DE proteins across the genetic-histological glioma subtypes and controls. The over-representation test was performed against the *Homo sapiens* reference database and a Fisher’s exact test was performed and corrected at a 5% FDR.

### 4.8. In Silico Analysis

Relative gene expression for plasma-EV proteins restricted to GBM/GIV patients were analysed in silico using TCGA gene expression data (Human Genome U133A Array, TCGA 2013) via the Oncomine^TM^ platform (Compendia Biosciences, MI, USA) [107]. Genes with significant transcript levels (Student’s *t*-test; *p*-value < 0.05) between GBM (*n* = 542) and normal brain tissue (*n* = 10) are presented as box-plots, generated in Excel.

### 4.9. Data Availability

The raw SWATH-MS acquisitions of the plasma-EV peptides and the SWATH spectral library are available in PeptideAtlas with the identifier PASS01487.

## 5. Conclusions

Future studies analysing plasma-EV proteins in larger longitudinal cohorts of glioma patients may substantiate the markers determined in this study for the conceptualisation of a glioma liquid biopsy. Testing of larger cohorts of glioma patients will improve the power of future studies and allow for machine learning methods to train, validate and test potential glioma plasma-EV biomarker panels. Furthermore, serial blood sampling and correlating plasma-EV profiles to clinical, pathological and radiological parameters will open avenues for building predictive models that stratify patients and devise clear biomarker thresholds for glioma progression and recurrence.

Using SWATH-MS, we achieved the most comprehensive and in-depth proteomic coverage of plasma-EVs to-date and established a clear advantage of applying this analytical platform to blood-based EV biomarker discovery. Putative plasma-EV biomarker proteins were identified between glioma genetic-histological subtypes and WHO grades, including those previously reported as markers of GBM invasion and aggression. Plasma-EV protein profiles reflect glioma subtypes and aggressiveness, and shows real promise for defining diagnostic, prognostic and predictive biomarker panels. Defining clinically relevant biomarkers that offer non-invasive, early indications of tumour recurrence are likely to have significant clinical utility, allowing accurate evaluation of glioma progression and treatment responses and in turn, improve the clinical management of patients.

## Figures and Tables

**Figure 1 ijms-21-04754-f001:**
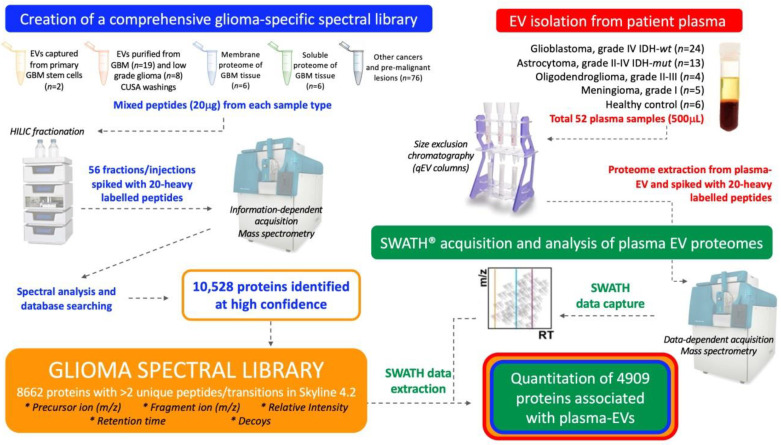
Experimental workflow for in-depth proteomic characterisation of plasma-extracellular vesicles (EVs). A custom spectral library was created by information dependent acquisition (IDA)-based LC-MS/MS of hydrophilic interaction liquid chromatography (HILIC) fractionated peptides derived from glioblastoma (GBM) specimens and other cancers. Database searching of 56 HILIC fractions identified 10,528 proteins. Proteins with more than two unique peptides and transitions were selected for the creation of a spectral library, comprised of 8662 protein species that contained reference sequences (precursor ion (*m/z*), fragment ion (*m/z*), relative intensity, retention time and decoys). Size exclusion chromatography (SEC) captured plasma-EVs and extracted proteomes were then assessed by sequential window acquisition of all theoretical fragment ion spectra mass spectrometry (SWATH-MS); protein data was extracted by aligning the SWATH-MS retention times to the spectral library using 20 heavy-labelled *PepCalMix* peptides that were spiked into both the plasma-EV and spectral library specimens.

**Figure 2 ijms-21-04754-f002:**
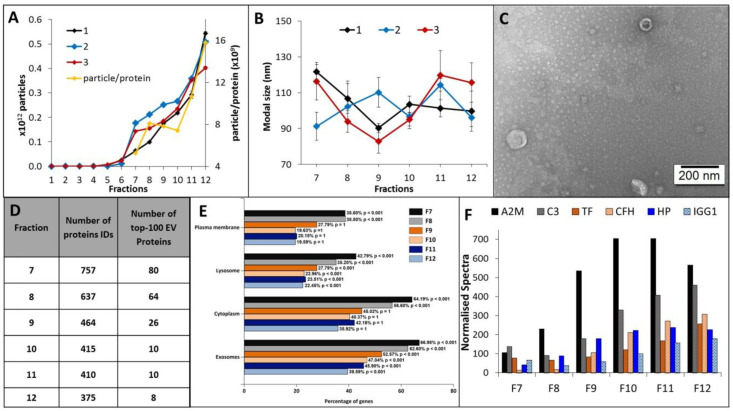
Characterisation and annotations of plasma-EV fractions. Nanoparticle tracking analysis (NTA) determined (**A**) number of particles and average particle to protein ratio, and (**B**) modal size distribution for 12 500-µL PBS EV fractions (F1-12) isolated from the plasma of three healthy individuals (1, 2, 3). The results are shown as the mean ± standard error of three independent NTA measurements. (**C**) Close-up TEM image of F7-9 EVs, 200 nm scale bar. (**D**) The number of confident protein identifications, including the top-100 EV-related proteins as reported by Vesiclepedia, determined by LC-MS/MS measurements of F7-12 EVs. (**E**) Functional enrichment annotations to cellular compartments for the LC-MS/MS sequenced F7-12 EVs. (**F**) Total normalised spectra for common serum proteins (A2M, C3, TF, CFH, HP and IGG1) sequenced by LC-MS/MS in F7-9 plasma-EVs.

**Figure 3 ijms-21-04754-f003:**
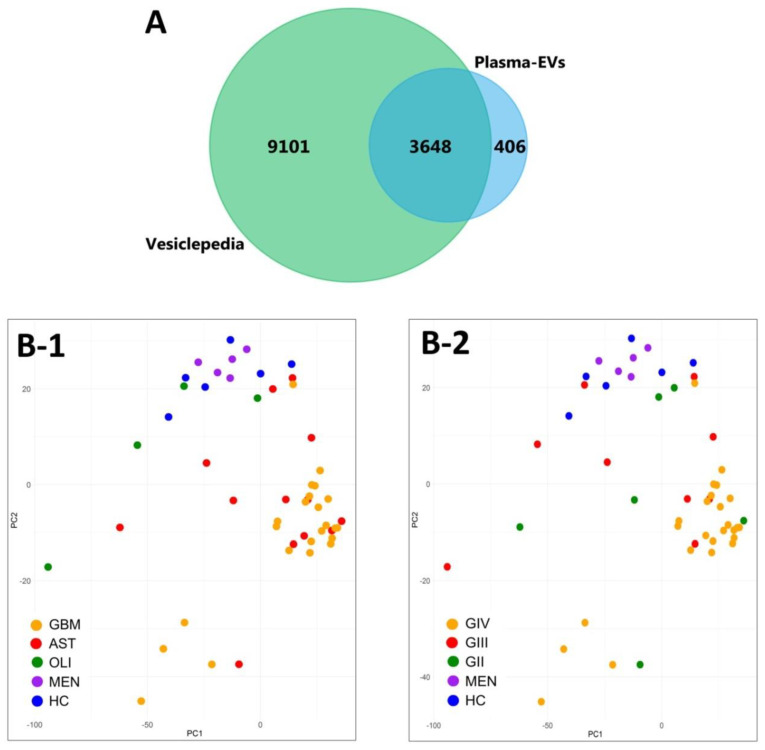
Characterisation of plasma-EV proteins by SWATH-MS. (**A**) Venn diagram showing overlap between the 4054 proteins confidently identified in the plasma-EVs in all sample groups by MSstats (blue) to EV proteins compiled in Vesiclepedia (green; matched by gene name; restricted to proteins annotated as originating from “extracellular vesicles”, “exosomes”, “microparticles” and “microvesicles”). (B) Unsupervised clustering by principal component analysis (PCA) based on the plasma-EV protein expression for samples grouped by their respective (**B-1**) genetic-histological subtype; glioblastoma (GBM; orange), astrocytoma (AST; red), oligodendroglioma (OLI; green), meningioma (MEN; purple) and healthy control (HC; blue); the proportion of the variance is explained by PC1 (*x*-axis) = 18.48% and PC2 (*y*-axis) = 7.28% (*y*-axis), and by (**B-2**) glioma grade; GIV (orange), GIII (red), GII (green), MEN (purple) and HC (blue); the proportion of the variance is explained by PC1 (*x*-axis) = 18.48% and PC2 (y-axis) = 7.28% (*y*-axis).

**Figure 4 ijms-21-04754-f004:**
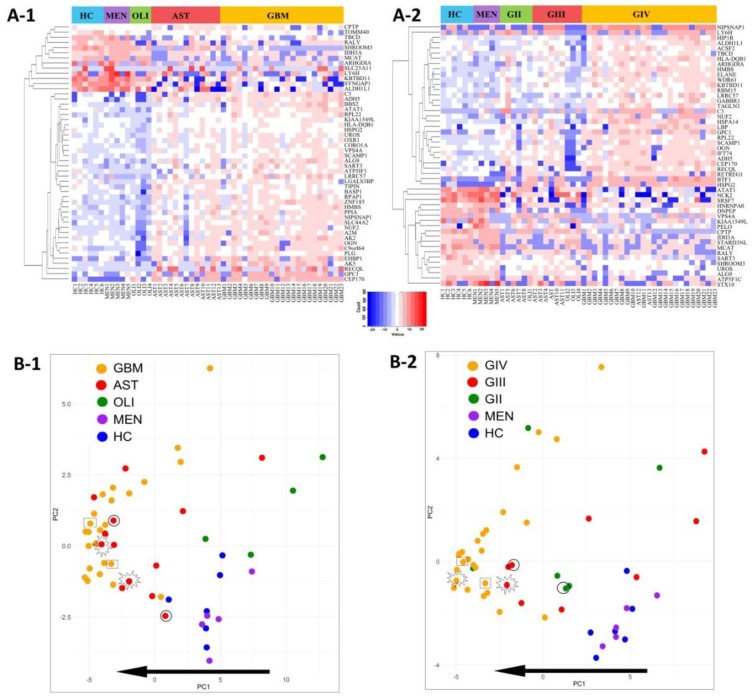
Visualisation of the top-50 differentially expressed (DE) proteins identified across glioma plasma-EVs. Heatmap representations of the top 50 significant DE plasma-EV proteins for samples categorised by (**A-1**) genetic-histological subtype and (**A-2**) WHO grade. Principal component analysis (PCA) based on the expression of the top-50 plasma-EV proteins by their respective (**B-1**) genetic histological subtype; HC (blue), MEN (purple), OLI (green), AST (red) and GBM (orange); the proportion of the variance is explained by PC1 (*x*-axis) = 42.95% and PC2 (*y*-axis) = 8.36% and (**B-2**) WHO grade; HC (blue), MEN (purple), GII (green), GIII (red) and GIV (orange); the proportion of the variance is explained by PC1 (*x*-axis) = 32.71% and PC2 (*y*-axis) = 11.49%. (B-1, B-2) Glioma samples that are circled were derived from an AST patient that progressed from GII to GIII. Samples that are starred were derived from an AST patient that progressed from GIII to GIV, while samples that are enclosed in a rectangle were derived from a recurrent GBM (GIV). The arrow depicts the direction of increasing glioma aggression.

**Figure 5 ijms-21-04754-f005:**
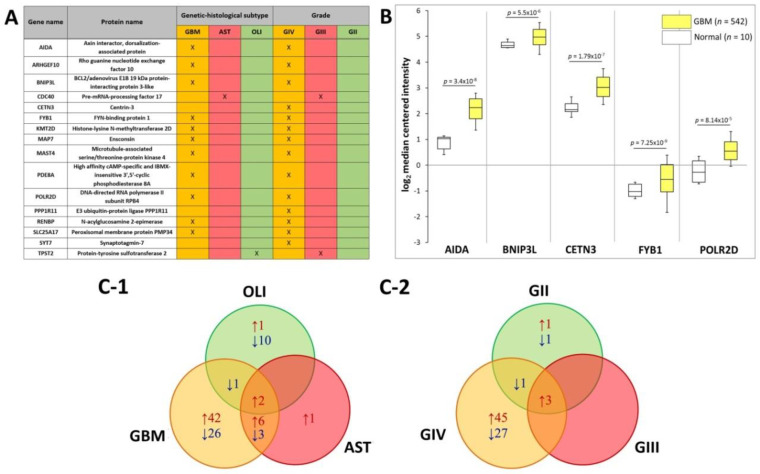
Putative plasma-EV protein markers for glioma. (**A**) Proteins with restricted expression in plasma-EVs according to different glioma genetic-histological subtypes (GBM, AST, OLI) and WHO grades (GIV, GIII, GII). Proteins that are present in the plasma-EVs of their respective glioma subtype are denoted by X. (**B**) Box-plots were generated by Oncomine for plasma-EV proteins restricted to GBM/GIV patients that revealed significant relative GBM gene expression levels (represented by log_2_ median centred intensity; U133A Array, TCGA 2013). Significant transcripts in GBM tissue included AIDA (2.5-fold, *p* = 3.4 × 10^−8^), BNIP3L (1.2-fold, *p* = 5.5 × 10^−6^), CETN3 (1.8-fold, *p* = 1.7 × 10^−7^), FYB1 (1.8-fold, *p* = 7.25 × 10^−9^) and POLR2D (1.7-fold, *p* = 8.14 × 10^−5^). *n* is the number of samples, the 90th percentile is signified by upper error bars, and the 10th percentile is represented by the lower error bars. (C) Plasma-EV protein abundance changes that were significant in glioma relative to both healthy and non-glioma controls (|fold-change| ≥ 2, *p* ≤ 0.05). The Venn diagrams show overlap of the glioma-related plasma-EV proteins in (**C-1**) GBM, AST and OLI and (**C-2**) GIV, GIII and GII plasma-EVs, where arrows denote direction of fold change relative to controls. The identities of glioma-related plasma-EV proteins are listed in Appendix A.

**Figure 6 ijms-21-04754-f006:**
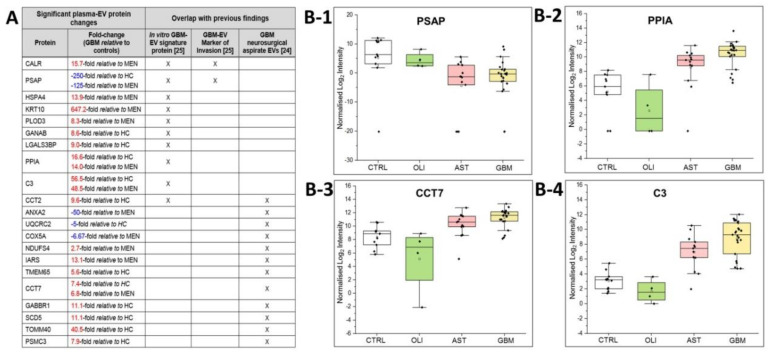
Overlap of significant GBM/GIV plasma-EV proteins with previous GBM-EV findings. (**A**) Significant, DE proteins in GBM plasma-EVs relative to either MEN or HC (|FC| ≥ 2, adj. *p* ≤ 0.05) also reported as in vitro GBM-EV signature proteins [29] and/or significantly expressed in EVs captured from glioma neurosurgical aspirates [28]; overlapping findings are denoted by ‘X’. Box-plots show the distribution of significant plasma-EV proteins (**B-1**) PSAP, (**B-2**) PPIA, (**B-3**) CCT7 and (**B-4**) C3. The box-plots depict the expression levels for CTRL (total controls, *n* = 11; including non-glioma, *n* = 5 and healthy controls, *n* = 6; white), OLI (*n* = 4; green), AST (*n* = 13; pink) and GBM (*n* = 23; yellow). The protein expression for each individual patient is plotted as normalised log2 intensity and denoted by a black dot. The upper error bars signify the 90th percentile, and lower error bars represent the 10th percentile, the middle line represents the median and open square signifies the mean.

**Table 1 ijms-21-04754-t001:** Overview of pre-operative blood specimen cohorts and grouping schema. Plasma-EV samples were grouped by their respective genetic-histological subtype and WHO grade.

***Genetic-Histological Subtype***				
**Cohort**	**Sample (*n*)**	**Gender**	**Age at Resection (Mean ± SD)**	**Histopathological Diagnosis**
GBM	24	17M/7F	60.8 ± 14.1	Primary Glioblastoma, *IDH-wildtype*, WHO grade IV
AST	13	10M/3F	40.1 ± 13.8	Astrocytoma, *IDH-mutant*; WHO grade II (*n* = 5), grade III (*n* = 6) and grade IV (*n* = 2)
OLI	4	4M	36.5 ± 3.7	Oligodendroglioma, *IDH-mutant*; WHO grade II (*n* = 1) and grade III (*n* = 3)
***WHO Grade***				
**Cohort**	**Sample (*n*)**	**Gender**	**Age at Resection (Mean ± SD)**	**Histopathological Diagnosis**
GIV	26	19M/7F	59.4 ± 14.6	WHO grade IV astrocytoma; *IDH-wildtype* (*n* = 24) and *IDH-mutant* Glioblastoma (*n* = 2)
GIII	9	8M/1F	39.8 ± 13.5	WHO grade III glioma; *IDH-mutant* astrocytoma and *IDH-mutant* oligodendroglioma
GII	6	4M/2F	37.3 ± 11.9	Grade II glioma; *IDH-mutant* astrocytoma and *IDH-mutant* oligodendroglioma
***Controls***				
**Cohort**	**Sample (*n*)**	**Gender**	**Age at Resection (Mean ± SD)**	**Histopathological Diagnosis**
MEN	5	3M/2F	57.6 ± 11.0	Meningioma, grade I (non-glioma controls)
HC	6	3M/3F	46.8 ± 7.4	Healthy controls

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
