# Peer review of "A Comprehensive Proteomic SWATH-MS Workflow for Profiling Blood Extracellular Vesicles: A New Avenue for Glioma Tumour Surveillance"

_ijms, 2020, doi:10.3390/ijms21134754_

Round 1

Reviewer 1 Report

The paper entitled " A comprehensive proteomic SWATH-MS workflow for profiling blood extracellular vesicles: a new avenue for glioma tumour surveillance” is interesting and exaustive in respect to the topic. State-of-art reflect major advances in the field. The paper is well written, and has merit of publication.

Author Response

No changes required.

Reviewer 2 Report

Review for ijms-830666

Work summary:

Susannah Hal et.al in this manuscript implemented an approach of proteomics called SWATH-MS (Sequential window acquisition of all theoretical fragment ion spectra mass spectrometry). This help determine low-abundant and rarely detected protein from plasma derived-EV from Glioma patient. Authors need to provide greater details of characterization on EVs and also validation of key identified protein from proteomics. Following are my comments.

Major comments:

1) iZON qEV SEC columns recommend 1-20 to determine the protein contamination of samples in later fraction (6-12 EV and 14-20 for free proteins). Indeed, focus of this study is EV that are known to elute in fraction 6-12, however, the purpose of characterization is important to determine all factions for protein amount. Based on latest MISEV2018 guideline (PMID 30637094) it is essential to each fraction to be determined for protein and particles. Ratio of particle/protein can be plotted for each fraction to give clear picture if earlier fraction is not contaminated with free proteins as this could happen.

2) In SEC the isolation of EV is based on size of particles it is likely be contaminated Lipoproteins (PMID 29441425). From EM picture it’s not clear If the image is of Lipoprotein or EVs. EVs usually are electron dense and dark with negative staining and have cup shape structure. Ideally, for such samples immune EM will be recommended to show some of EV enriched protein associated with EVs.

Recommended experiments can be done in controls subjects as the material will be limited.

3)   Validation of identified proteins is essential to validate the finding from proteomic study. I can image the samples of will be limiting factor. Antibody based detecting of identified membrane proteins could be used either by ELISA (PMID: 31595182) or Western. Validation is essential for authenticating the findings. For example, mitochondrial protein (in this study TOMM40) was also found in previous study could be interrogated (PMID:  31497264)

Minor comments

1) Some analysis author use Vesiclepedia and in other they used Exocarta. Could author use Vesiclepedia as its latest set protein.  

2) Could author also perform comparative analysis identified from their data with transmembrane protein present Vesiclepedia. These will be reliable candidate for validations.

3) Could author discuss drawback of this method over other technique.

4) Introduction could be shortened.

Author Response

Major comments:

1) We understand that the Izon Automated Fraction Collector manual shows profiles for fractions 1-20 for qEV70 and qEV35 columns, however, the current qEV manual recommends collecting 12 x 500uL fractions (where fractions 7-9 contain EVs of highest purity). We have used qEV SEC columns to isolate blood EVs for previous studies (PMID 29084979 and 30564636) and routinely collect 12 x 500uL fractions.

In accordance with MISEV2018 guidelines we have provided two quantitative methods (Nanosight tracking analysis and Total protein quants by Qubit fluorometric assay) for the twelve fractions collected. We have added the particle to protein ratio for the EV containing fractions (F7-12) in Figure 1A.

2) We agree that immune EM or cryo-EM would provide clearer images of the EV lipid bilayer than conventional TEM and can better distinguish plasma-EVs from lipoprotein contaminants. We only had access to TEM at the time of this study, and COVID restrictions to our University facilities are currently in place so any additional experiments are not possible in the foreseeable future. We have selected a perhaps, better image that shows the cup-shaped morphological artifact attributed to exosomes.

3) SWATH mass spectrometry is a highly sensitive method and stringent quality control and thresholds were set for protein identification and quantification. Sample availability and volume are indeed limiting factors for future validation work. We agree that further studies are necessary to authenticate the many interesting putative biomarkers identified here and have added the following to the discussion (lines 422-4, page 13):

“The development of assays using targeted DIA strategies (i.e., selective/parallel reaction monitoring) or immunoassayswill be necessary to validate the putative glioma EV markers resolved here.”

However, such studies are beyond the scope of the current manuscript, which aims to establish a method to allow comprehensive proteomic profiling of circulating-EVs and further determine whether this approach can distinguish glioma subtypes and control cohorts.

Minor comments

1) We have amended this. ‘Vesiclepedia-listed’ top-100 EV proteins are now reported in Figure 2D and Supplementary TableS6. The figure legend describes figure 2D (page 5, line 206) and text (page 5, line 191) has also been edited.

2) Figure 2E shows the overlap of EV proteins with plasma membrane proteins and other membranous compartments annotated in Vesiclepedia. In addition, the expression of multiple tetraspanin proteins including CD63, CD81 and ITGB1, are listed as top-100 EV proteins in Supplementary TableS6.

3) We have added the following to the discussion (page 13, line 460-464). "While SWATH-MS sequences all peptides in a sample, in-depth protein identification is constrained by the extensiveness of the spectral library used. As spectral libraries mature, plasma-EV samples can be re-analysed to achieve more complete protein profiles, however, analyses of large SWATH datasets are both time and computationally intensive”.  

4) While we agree that the introduction is quite lengthy, we have concisely introduced multiple concepts (glioma and current clinical challenges, extracellular vesicles, blood proteomics and drawbacks of shotgun mass spectrometry, and SWATH-MS) that are integral to understanding the aims of this study.

Round 2

Reviewer 2 Report

Round 2 _Reviewer Response (RR)

2) We agree that immune EM or cryo-EM would provide clearer images of the EV lipid bilayer than conventional TEM and can better distinguish plasma-EVs from lipoprotein contaminants. We only had access to TEM at the time of this study, and COVID restrictions to our University facilities are currently in place so any additional experiments are not possible in the foreseeable future. We have selected a perhaps, better image that shows the cup-shaped morphological artifact attributed to exosomes.

RR 2: I agree with current difficulty to perform EM, hence presented EM is acceptable. However, considering the samples type it would ideal to include the analysis the data in the light identified transmembrane proteins in fig 2D and discuss the identified transmembrane proteins  

3) SWATH mass spectrometry is a highly sensitive method and stringent quality control and thresholds were set for protein identification and quantification. Sample availability and volume are indeed limiting factors for future validation work. We agree that further studies are necessary to authenticate the many interesting putative biomarkers identified here and have added the following to the discussion (lines 422-4, page 13):

“The development of assays using targeted DIA strategies (i.e., selective/parallel reaction monitoring) or immunoassays will be necessary to validate the putative glioma EV markers resolved here.”

However, such studies are beyond the scope of the current manuscript, which aims to establish a method to allow comprehensive proteomic profiling of circulating-EVs and further determine whether this approach can distinguish glioma subtypes and control cohorts.

RR 3: Exactly, since the method is sensitive and hence validation of not all but few proteins especially transmembrane protein would help the quality of manuscript and method. Any method could be used to give a hint that claimed proteins are really present.

Author Response

RR2. It is not clear what data analysis the Reviewer is referring to or why (specifically) identified transmembrane proteins should be discussed.

The purpose of Fig 2D is to communicate the number of confident protein IDs and number of Top-100 EV proteins identified in each EV fraction. This data is available in Table S6. Fig 2E then shows the enrichment of significant cell compartment (annotations include 'plasma membrane') for each fraction. This information was used to select the most appropriate fractions (i.e., 7-9) for downstream EV SWATH analysis. 

RR3. While proteins are traditionally confirmed by immunoblotting, LC-MS/MS was preferred in this study as it is A) a sensitive method for which minimal EV sample is sacrificed, B) not dependent on the efficiencies of antibodies and C) according to MISEV guidelines, is sufficient for EV protein characterisation.

The Reviewer recommends that "any method could be used to give a hint that claimed proteins are really present." 

Our manuscript presents data from two different mass spectrometry and data processing methods, IDA by a Q-ExactiveTM Plus and DIA (SWATH) by a TripleTOF® 6600. Multiple EV-related and glioma-associated proteins were detected in the plasma-EVs by both MS methods.

We have amended Table S7A, which now displays the top-100 EV proteins (as listed by Vesiclepedia) and overlapping EV-related proteins that were detected by both IDA and SWATH-MS analyses. 

We have added the following to the text:

"Multiple classical EV proteins (68 proteins, including, ACTB, ACTN1/4, ANXA1, ANXA2, ANXA6, CD9, FLOT1, GAPDH, HIST1H4A, HSP90AB1, ITGA6, ITGA2B, PDCD6IP, SLC3A2; Table S7A) and glioma-associated proteins (including CCT7, PPIA and C3; Table S6 and Table S7A) were detected in the plasma-EVs by both the IDA and SWATH (DIA) MS methods." (lines 423-7).